# In Vitro and In Silico Analysis of Epithelial-Mesenchymal Transition and Cancer Stemness as Prognostic Markers of Clear Cell Renal Cell Carcinoma

**DOI:** 10.3390/cancers15092586

**Published:** 2023-05-01

**Authors:** Revati Sharma, Showan Balta, Ali Raza, Ruth M. Escalona, George Kannourakis, Prashanth Prithviraj, Nuzhat Ahmed

**Affiliations:** 1Fiona Elsey Cancer Research Institute, Ballarat, VIC 3353, Australia; revatisharma@students.federation.edu.au (R.S.); alir@students.federation.edu.au (A.R.); ruth@fecri.org.au (R.M.E.); george@fecri.org.au (G.K.); prashanth@fecri.org.au (P.P.); 2Health Innovation and Transformation Centre, Mt Helen Campus, Federation University Australia, Ballarat, VIC 3350, Australia; 3Dorevitch Pathology, Ballarat Base Hospital, Drummond Street, Ballarat, VIC 3350, Australia; showan.balta@dorevitch.com.au; 4Centre for Reproductive Health, The Hudson Institute of Medical Research, 27–31 Wright Street, Clayton, VIC 3168, Australia; 5Department of Molecular and Translational Science, Monash University, Clayton, VIC 3168, Australia; 6Department of Obstetrics and Gynaecology, University of Melbourne, Parkville, VIC 3010, Australia

**Keywords:** clear cell renal cell carcinoma, epithelial-mesenchymal transition, cancer stem cells, prognostic markers

## Abstract

**Simple Summary:**

Clear cell renal cell carcinoma (ccRCC) is a highly aggressive cancer responsible for about 85% of kidney cancers. The majority of the incidentally detected renal masses are small and confined to the kidney; however, a significant number of patients initially present with progressive metastatic cancer succumb to the disease in a short time frame. High levels of expression of hypoxia-inducible transcription factors (HIF) resulting in the downstream epithelial-mesenchymal transition (EMT) pathway and features of cancer stem cells (CSCs) leads to an aggressive and drug-resistant phenotype in ccRCC. In this paper, using data from in-house collected patient tumours and public domain datasets, we highlight EMT and CSC to be prominent players in ccRCC progression. Using these approaches of analysis, we show the development of multi-marker diagnostic and prognostic signatures, which may stratify high-risk patients likely to progress to metastatic disease.

**Abstract:**

The process of epithelial-mesenchymal transition (EMT) involves the phenotypic transformation of cells from epithelial to mesenchymal status. The cells exhibiting EMT contain features of cancer stem cells (CSC), and the dual processes are responsible for progressive cancers. Activation of hypoxia-inducible factors (HIF) is fundamental to the pathogenesis of clear cell renal cell carcinoma (ccRCC), and their role in promoting EMT and CSCs is crucial for ccRCC tumour cell survival, disease progression, and metastatic spread. In this study, we explored the status of HIF genes and their downstream targets, EMT and CSC markers, by immunohistochemistry on in-house accrued ccRCC biopsies and adjacent non-tumorous tissues from patients undergoing partial or radical nephrectomy. In combination, we comprehensively analysed the expression of HIF genes and its downstream EMT and CSC-associated targets relevant to ccRCC by using publicly available datasets, the cancer genome atlas (TCGA) and the clinical proteome tumour analysis consortium (CPTAC). The aim was to search for novel biological prognostic markers that can stratify high-risk patients likely to experience metastatic disease. Using the above two approaches, we report the development of novel gene signatures that may help to identify patients at a high risk of developing metastatic and progressive disease.

## 1. Introduction

Clear cell carcinoma (ccRCC) is a heterogeneous tumour distinguished by the manifestation of symptoms of the advanced-stage disease at diagnosis. Although improving, the survival rate of advanced ccRCC is still low (Stage I–IV—88%, 63%, 65% and 15%, respectively), with 12% of the diagnosed cancer progressing to the metastatic stage [1]. Accurate data on the particular risk of cancer progression and death at diagnosis and after treatment is essential to guide patients, plan personalised investigation procedures, and select patients for targeted treatment protocols and new clinical trials. Tumour markers, present in tumours and biological fluids of the tumour-bearing host, can have diagnostic, prognostic, predictive, and therapeutic potential, and their levels are likely to change from normal with cancer progression. Predictive biomarkers can predict whether a patient will have a positive or negative outcome with a particular treatment [2]. In contrast, diagnostic and prognostic markers can help detect the disease and measure its progression in patients. Identifying specific gene signatures (biological markers) that can detect patients prone to metastatic disease can help in designing treatment strategies that can slow the progression of the disease, resulting in prolonged progression-free survival [3].

Hypoxia is one of the major contributors to ccRCC progression [4]. Hypoxia, through the activation of hypoxia-inducible factors (HIFs), inflicts adaptive changes within cancer cells, resulting in aggressive behaviour leading to tumour progression and treatment resistance, which consequently leads to unfavourable prognosis in patients [4]. In healthy renal epithelial cells, the expression of von Hippel–Lindau tumour suppressor protein (pVHL)/E3 ubiquitin ligase complex leads to the proteasomal degradation of HIF-1α and HIF-2a subunits, which ensures that HIF-1α and HIF-2α are inactivated [5]. However, VHL/E3 ubiquitin ligase is inactivated in the majority of the ccRCC; as a result, HIF-1α and HIF-2α cannot be degraded and, therefore, are constitutively expressed in the majority of ccRCCs [5]. Both HIF-1α and HIF-2α share structural and functional similarities but have markedly different target genes. HIF-1α is known for tumour suppressor functions, and HIF-2α has been reported to promote tumorigenicity in ccRCC [6,7].

EMT has been heralded as a hallmark of ccRCC progression [8,9]. The process is initiated by various tumour microenvironmental (TME) stimuli that cancer cells receive, one of which is hypoxia-initiated HIF activation, which is instrumental in the commencement and promulgation of EMT [10,11]. Recent studies have revealed oncogenic and tumour-suppressive roles of HIF proteins in ccRCC instigation and propose that modification in the equilibrium of HIF-1α and HIF-2α activities can have a role in ccRCC progression and aggressiveness [12]. Hypoxic activation of HIFs has been shown to contribute to tumour aggressiveness by stimulating multiple molecular pathways, including the EMT and CSCs, resulting in poor prognosis [10]. A recent study based on the extrapolation of TCGA data on 533 ccRCC patients showed the classical epithelial marker E-cadherin (E-cad) to be decreased, while the expression of mesenchymal markers N-cadherin (N-cad), SNAI1, vimentin (VIM), TWIST1 to be increased in ccRCC primary tumours, compared to control normal tissues [13]. This study further showed that the patients with high expression of VIM, TWIST or low expression of E-cad had worse prognostic outcomes. Cox regression analysis suggested that EMT markers (E-cad, SNAI1, VIM, and TWIST) were independent prognostic factors of both progression-free survival (PFS) and overall survival (OS) in ccRCC patients [13]. In another study, patients diagnosed with ccRCC showed higher expression of ZEB2 and TWIST proteins in low-grade compared to high-grade tumours, which may imply that the process of EMT is initiated at the early stages of tumour development, suggesting that the evaluation of EMT-associated proteins, may be useful for the assessment of the metastatic potential of tumours in patients [14]. In that scenario, the dissemination of ccRCC cancer cells leading to a progressive disease can be promoted by the deficit of epithelial cell adhesion molecule E-cad and upregulation of E-cad repressors such as Slug, Snail, ZEB, and TWIST, which are the hallmarks of the EMT process [8]. The above studies suggest that screening for EMT markers in ccRCC tumours may provide prognostic evaluation for the patient’s risk of developing progressive disease.

EMT inducer such as transforming growth factor β1 (TGFβ1) has been shown to induce a variety of complex signalling pathways in cancer cells [11,15]. Along with the regular EMT markers that have been studied vigorously in cancers, the roles of other non-classical genes have been linked with EMT in cancers. In that context, Aldo-keto reductase family 1, member B1 (AKR1B1), can induce EMT via a positive feedback loop between TWIST and NF-κB [16]. Integrin subunit αv (ITGAV) has been identified as an EMT marker in breast cancer and is associated with tumour cell detachment leading to metastatic spread of the tumour [17,18]. AKR1B1 and ITGAV have been associated with drug resistance associated EMT in cancer [19]. Alpha smooth muscle actin (α-SMA), another cytoskeletal marker like VIM, has been known to contribute to EMT in cancer [20]. In addition, glucose-6 phosphate dehydrogenase (G6PD) regulates EMT and metabolic reprogramming in cancer cells [19].

The coalition of EMT with CSC-like cells has been demonstrated in many cancers [21,22]. In that context, stimulation by TGFβ1 results in both EMT and CSC-like cells [23]. Consistent with this, the association of mesenchymal markers TWIST or SNAI1, accountable for the inhibition of the epithelial adhesion molecule E-cad, has been shown to stimulate CSC-like cells in breast cancer [22]. E-cad transcriptional repressors Snail and SNAI1(Slug) induce invasiveness and CSC-like features and chemoresistance in ovarian cancer cells [24,25], suggesting that CSCs can be tackled by pharmacological inhibition of transcription factors that induce EMT in cancer [26]. Like most cancers, no generally relevant group of CSC markers has been identified in ccRCC. The characterisation of known CSCs varies in individual studies and is justified by their functional characteristics [27]. Several markers were found to be specifically expressed in CSCs derived from ccRCC, some of which are CD44, CD105, ALDH1, OCT4, CD133, and CXCR4 [27,28]. A recent study has shown the existence of CD133 expressing CSCs in ccRCC tumours [29]. In addition, exosomes from CD103^+^ expressing CSCs were shown to be increased in blood samples from ccRCC patients with lung metastasis [30]. These CSC-based exosomes promoted metastasis to the lungs in a mouse model of ccRCC. Recent studies have identified transcription factors EZH2 [31] and CD73 [32] as putative CSCs in RCC with enhanced metastatic abilities in cell lines and mouse models. In another study, CSCs derived from sphere-forming assays using ccRCC cell lines were shown to be enriched in IL-8 and CXCR1 expression [33], indicating the association of IL-8/CXCR1 with CSC-like properties in ccRCC. In addition, tumour-infiltrating macrophages secrete inflammatory factors and cytokines, such as TGFβ, IL-6, IL-10, and tumour necrosis factor α (TNFα), which promote EMT, have been shown to enrich tumours with CSC-like properties [34]. These observations suggest a putative role of cancer cells expressing different EMT and CSC markers in ccRCC progression.

Although EMT and CSC play an important role and impact the progression of ccRCC in different ways, the interplay of these processes in terms of diagnostic and prognostic significance is not clearly defined in ccRCC. In this study, we have selected and comprehensively evaluated the expression of four commonly known EMT markers (E-cad, N-cad, VIM, TGFβ1), and four unconventional EMT markers (α-SMA, AKR1B1, ITAV, G6PD), less studied in cancer in relation to the EMT process. The idea was to test the potential of these unconventional EMT markers in the progression of ccRCC. In addition, we studied the known CSC markers in ccRCC (CD44, CD133, and CD105), by in vitro immunohistochemistry staining and extrapolation of publicly available datasets like TCGA and CPTAC, with the goal of contributing to the search for novel molecular ccRCC signatures which may lead to diagnostic/prognostic evaluation of patients. Most high-risk ccRCC patients undergo clinical observation for tumour growth after initial diagnosis, which in many cases can continue for years without any intervention. In that scenario, this patient group may miss the window of opportunity for therapeutic treatment at an early stage, which ideally would prolong progression-free survival in that group. This study thus addresses that scenario and provides an initial framework for clinicians to assess high-risk patients at an early stage to implement specific disease management plans to prolong disease-free survival.

## 2. Materials and Methods

### 2.1. RCC Patient Samples

#### 2.1.1. Ethics Approval

The Human Research Ethics Committee of Ballarat Health Services (BHS) and St John of God Ballarat Hospital approved the use of patient tissue samples (Project ID: 37521).

#### 2.1.2. Archived Patient Tissues

Tumours from ccRCC patients used in this study were formalin-fixed and paraffin-embedded. They were obtained from two sources, the Department of Pathology, Ballarat Base Hospital (BHS) and Hunter Cancer Biobank (HCB), NSW Regional Biospecimen and Research Services. To access the archived FFPE tissue from the BHS, informed consent was obtained from each patient. For the HCB, relevant permissions were obtained from their Research Review committee. A total of 19 FFPE tissues were obtained from the above-mentioned sources. Fourteen patient tissue paraffin blocks, along with the history of the patients, were obtained from BHS, and 5 patient tissues were obtained without any history from HCB. Out of the total 19 tumour tissues, 13 were primary tumours, and six were metastatic. Table 1 enlists the description of patients who participated in the study. Throughout the study, the patient data was handled anonymously by assigning a number to each patient.

#### 2.1.3. Collection of Clinical Data

Clinical and pathological data were retrospectively collected by viewing the patient’s pathological reports. The following variables were collected when available, age, sex, tumour size and grade, and date of nephrectomy or biopsy. Although the sections from Hunter Biobank (all metastatic tumours) did not come with relevant patient history and were unknown, the stages of the tumour sections were confirmed by a pathologist, Showan Balta, Dorevitch Laboratories, Ballarat Base Hospital. The grade could not be confirmed as there was no patient history available for these tissues. Balta also helped in identifying normal adjacent kidney tissues in primary and metastatic ccRCC tumours.

#### 2.1.4. Immunohistochemistry and Pathological Evaluation of In-House Collected Tissues

Immunohistochemistry (IHC) was performed by a standard pathology procedure. In brief, IHC was performed on 4 μm paraffin-embedded sections, which were deparaffinized, followed by antigen-retrieval in citrate/EDTA buffer. This was followed by the treatment of slides with optimum concentration of primary antibodies at 4 °C overnight. Following a few phosphate-buffered salines with tween 20 (PBST) washes, the slides were treated for 60 min with secondary rabbit/mouse horseradish peroxidase antibodies (Agilent, Dako, Sydney, Australia). After PBST washes, slides were incubated for 5 min with the EnVisionTM FLEX DAB (3,3′-Diaminobenzidine) chromogen + substrate (Agilent Dako, Sydney, Australia), stained with Mayer’s haematoxylin (Sigma-Aldrich, St. Louis, Missouri, United States) for 5 s, and rinsed with tap water. The sections were dehydrated, covered with coverslips, and air-dried. IHC slides were scanned using an EVOS microscope, and the images were exported to a commercially available digital imaging and semi-quantification software Aperio ImageScope v10 [35,36,37,38]. The algorithm parameters were customised to differentiate between negative (blue), weak (yellow), moderate (orange), and strong (red) stained cells (Appendix A). The data is presented as the number and intensity of the positive pixels.

### 2.2. Analysis of Publicly Available RCC Datasets Using Interactive Web Tools

#### 2.2.1. OncoPrint Analysis of EMT and Hypoxic Markers Using cBioPortal

The cBioPortal for cancer genomics (http://www.cbioportal.org) (accessed on 8 January 2023) is a web portal for exploring genetic alterations across samples, genes, and pathways. The current study used the cBioPortal OncoPrint for ccRCC patient samples (*n* = 538) to acquire a precise graphical summation of gene expression alterations in selected genes. Publicly available Kidney ccRCC TCGA (The Cancer Genome Atlas) datasets (TCGA-Firehose legacy *n* = 538) were used to query the gene alterations in 11 genes in Kidney ccRCC (TCGA Provisional) case set.

The genes selected were CDH1 (E-cad), CDH2 (N-cad), VIM, ACTA-2 (α-smooth actin, α-SMA), TGFB1 (TGFβ1), ITGAV, AKR1B1, G6PD, ENG (CD105), CD44 and Prom 1 (CD133). Genomic alterations, including copy number alterations (CNAs) (amplifications and homozygous deletions), mutations, and alterations, including missense truncating, in frame and fusion genes, were examined in the coding sequence of each gene. Glyphs and colour coding summarise the gene or protein expressions. All cases are arranged as per alterations.

#### 2.2.2. Gene and Protein Expression across Normal and ccRCC Samples Using UALCAN

The UALCAN (http://ualcan.path.uab.edu) (accessed on 17 March 2023) web portal is a user-friendly interactive tool that can perform in-depth analyses of the TCGA and CPTAC data. We used the CPTAC database (KIRC) on the UALCAN portal to analyse the proteomic expression profiles of selected proteins in the ccRCC tumours and their adjacent normal kidney tissue based on clinicopathological parameters of the cancer stage.

#### 2.2.3. Survival Curves Using the GEPIA Web Tool

Gene expression profiling interactive analysis (GEPIA) (http://gepia.cancer-pku.cn) (accessed on 26 April 2022) stores RNA sequencing data of 9736 tumours and 8587 normal kidney tissues from the TCGA and Genotype-Tissue Expression (GTEx) project using a standard processing procedure.

#### 2.2.4. Kaplan-Meier Curves Using PROGgeneV2 Web Tool

PROGgeneV2 (http://www.progtools.net/gene/) (accessed on 16 January 2023) is a web-based podium used to study prognostic associations of genes in different kinds of cancer using high throughput genomic data. PROGgeneV2 was specifically utilised in the current study to achieve survival analysis on a combination of genes as a signature in ccRCC.

## 3. Results

Thirteen markers (including upstream HIF-1a and HIF-2a) were investigated in this study using in vitro (IHC) and bioinformatics tools (TCGA and CPTAC). Table 2 enlists the gene and protein nomenclature of the markers studied with relevant references.

### 3.1. Oncoprint of the 11 Selected Genes by RNA Sequencing Using cBioPortal

The study investigated the genetic alteration of a panel of 11 selected genes in ccRCC patient tumours using oncoprint analyses from cBioPortal. The genes involved in EMT and CSC processes were shortlisted for reasons mentioned in the introduction of this paper [8]. Table 3 enlists the selected in vitro and in vivo tools used to study the markers described above in Table 2.

A concise graphical summary of genetic alterations within a set of ccRCC tumour samples can be viewed using an OncoPrint derived from the TCGA Firehose dataset (Figure 1). It shows the percentage of the ccRCC patient population that carried altered genes. The different types of genetic alterations that were found associated with ccRCC patients were missense mutations, splice mutations, truncating mutations, amplification, deep deletion, and low and high mRNA expression. The OncoPrint analysis of tumour tissues from ccRCC patients revealed that 33% of the patient tissues had an alteration in the queried genes. The most common alteration found in the examined gene panel was mRNA downregulation (139 cases) and mRNA upregulation (80 cases) (Figure 1).

A classic EMT trend was anticipated in most patients with the downregulation of the expression of epithelial markers like CDH1 and the upregulation of mesenchymal markers like CDH2 and VIM. However, according to the oncoprint analysis, all three markers, CDH1, CDH2 and VIM, also showed downregulated mRNA expression in some ccRCC samples. Around 4% of the patients had each of the three gene alterations. The investigated genes had multiple types of genetic alterations, with ITGAV (9%) bearing the highest number of mutations. All the mutations in ITGAV were variations of unknown significance (VUS), a genetic change whose impact on cancer risk is not yet understood.

A missense mutation was found in 7 cases, with mutations in CDH2 (3 cases), CDH1 (1 case), (VIM) (1 case), CD44 (1 case), and G6PD (2 cases). Splice mutations were found in CD44. A single case of truncating mutation, which was a putative driver, was found in CDH1. Truncating mutations (VUS) were identified in CDH2 (3 cases), ITGAV (17 cases), and AKR1B1 (3 cases). Twelve cases of gene amplification were detected and were rampant in ITGAV (5 cases), AKR1B1 (3 cases), G6PD (2 cases), CDH2 (1 case) and TGFB1 (1 case). Three deep deletions were found in the analysis distributed in CDH2 (2 cases) (Figure 1).

The above data suggests significant dysregulation of several genes associated with EMT and CSC in ccRCC. The data also suggest that a combination of approaches for specific targeting of these pathways may be required for successful treatment outcomes in ccRCC patients.

### 3.2. Expression of EMT Proteins in ccRCC Patient Tissues

As mentioned before, this study investigated the expression of several conventional and unconventional EMT markers. We adopted an integrative approach of data mining from the publicly available dataset TCGA and CPTAC in parallel to the expression of proteins in patient tumours collected in-house, using the immunohistochemistry (IHC) process. The in-house patient group consisted of tumours collected from 19 stages I–IV patients. mRNA and protein expression levels of different markers on ccRCC were compared using data accrued from in-house IHC staining and publicly available data sets. The protein expression of the markers in the adjacent normal kidneys (*n* = 84) and primary ccRCC (*n* = 110) was compared using publicly available CPTAC samples on the UALCAN database. The primary and metastatic in-house tumours (non-matching and matching groups) were compared for the protein expression levels of the different markers. Expression levels of markers were also compared amongst ccRCC stages and grades using CPTAC samples in the UALCAN database and in-house tissues, respectively. This was done to relate the expression of each marker in terms of tumour spread and aggressiveness. The overall survival of the patients was also evaluated based on the mRNA transcription levels of the various markers using the TCGA dataset, Kidney Renal Cell Carcinoma (KIRC) and GEPIA web tool.

#### 3.2.1. Expression of HIF-1α or HIF-2α in ccRCC

A representative IHC expression of HIF-1α in normal kidney tissue and adjoining ccRCC tumour is presented in Figure 2A. A comparison of the in-house patient’s normal kidney tissue by IHC showed the expression of HIF-1α to be moderate in the normal kidney tubules, while no staining in the primary tumour was observed (Figure 2A). On the other hand, metastatic ccRCC displayed strong HIF-1α staining compared to the primary tumour (Figure 2A). Contrary to that, in the CPTAC samples of the UALCAN database, an insignificant upregulation of HIF-1α protein expression was noted in primary tumours compared to normal kidney tissues (Figure 2B), which showed a similar trend in the two in-house matching patient’s primary and metastatic tumours. An enhanced expression of HIF-1α in the metastatic compared to the primary tumours was observed (Figure 2C,D). The pair of the matching primary tumour samples were grade 2 ccRCC, while the metastatic tumours were grade 3 ccRCC (Figure 2D).

The grades (in-house collected samples) showed a trend of increased HIF-1α expression in grades 2, 3 and 4 compared to grade 1 (Figure 3A). A similar trend was observed in cases of stages CPTAC samples, where a consistently enhanced HIF-1α expression was noted in stages 1,2, 3 and 4 compared to normal kidney tissues (Figure 3B). The prognostic value of HIF-1α was also analysed using the Kaplan-Meier survival plot and GEPIA database (Figure 3C). No prognostic significance of HIF-1α expression was noted on the OS of patients (Figure 3C).

On the other hand, the staining pattern of HIF-2α was completely different in normal kidney tissue, primary and metastatic ccRCCs compared to HIF-1α expression (Figure 2A). Moderate to strong expression of HIF-2α was noted in normal kidney epithelial cells, which persisted in primary and metastatic tumours (Figure 2A). Consistent with that, in the CPTAC samples, an insignificant upregulation of HIF-2α protein expression was noted in primary tumours compared to normal kidney tissues (Figure 2B), and that trend of staining was retained in the in-house patient’s primary and metastatic tumours and also in the matching primary and metastatic tumours which showed an enhanced expression of HIF-2α in the metastatic compared to the primary tumours (Figure 2C,D). The grades (in-house collected samples) and stages (CPTAC samples) showed a consistent trend of increased HIF-2α expression in grades 2, 3 and 4 compared to grade 1 (Figure 3A) and in stages 1, 2, 3 and 4 compared to normal kidney tissues (Figure 3B). The prognostic value of HIF-2α analysed using the Kaplan-Meier survival plot showed significance in the OS of patients when HIF-2α was expressed at a higher level (Figure 3C).

#### 3.2.2. Conventional EMT Markers E-Cad, N-Cad and VIM

E-cad, N-cad and VIM are considered the canonical markers of EMT [14]. Expression levels of these three classical EMT markers were compared between adjacent normal tissues and ccRCC using in-house accrued patient samples and publicly available databases.

The representative IHC expression of E-cad in normal adjacent kidney tissue and RCC tumour can be seen in Figure 4A. A comparison of the in-house patient’s normal kidney tissue compared to primary and metastatic tumours by IHC showed the expression of E-cad to be strong in the tubules of normal kidney tissue, slightly diffuse staining in the primary, with no staining in metastatic tumours (Figure 4A).

Consistent with that, in the CPTAC samples of the UALCAN database, a significant upregulation of E-cad protein was noted in the normal kidney tissues compared to the primary tumours (Figure 4B). However, a comparison of the in-house patient’s primary and metastatic tumours by IHC showed the expression of E-cad to be slightly upregulated in the primary compared to the metastatic tumours (Figure 4C). The study also compared two matching pairs of primary and metastatic tumours for E-cad expression (Figure 4D). Compared to primary tumours, the pairs showed a clearly reduced expression in the matching metastatic tumours.

As mentioned before, the differences in the expression of E-cad between matching primary and metastatic tumours may also be due to differences in the grades of primary and metastatic tumours. The pair of primary tumour samples were grade 2 ccRCC, while the metastatic tumours were grade 3 ccRCC. The grades (in-house collected samples) showed a trend of increased E-cad expression in grades 2, 3 and 4 compared to grade 1 (Figure 5A). In cases of stages, significantly decreased E-cad expression was noted in stages 1, 2, 3 and 4 compared to normal kidney tissues (Figure 5B). The prognostic value of E-cad was also analysed using the Kaplan-Meier survival plot and GEPIA database (Figure 5C). Low expression levels of E-cad were significantly associated with poor OS.

Like the E-cad, the expression level of N-cad was also investigated amongst all the above-mentioned ccRCC groups. Figure 4A shows a representative IHC expression of N-cad in primary and metastatic RCC tumours and adjacent normal tissue. Moderate expression of N-cad was noted in normal kidney tissue, with strong expression in the primary tumour and moderate membranous expression in the metastatic tumour (Figure 4A). In the CPTAC dataset, the normal kidney tissues expressed a significantly higher level of N-cad than the primary tumours (Figure 4B). Next, the in-house patient’s primary and metastatic tumours were compared for their N-cad expression levels using IHC. There was an insignificant elevated expression of N-cad in metastatic compared to the primary tumours (Figure 4C). Consistent with that, the two matching primary and metastatic tumour pairs showed an increased expression in the metastatic compared to primary tumours (Figure 4D). As mentioned above, for E-cad expression, the pair with the higher expression of N-cad was the metastatic grade 3 ccRCC, and the tumour pair with the lower expression were grade 2 primary tumours. The grades and stages of ccRCCs were screened for the expression levels of N-cad using IHC and publicly available CPTAC samples. In the CPTAC samples, the expression of N-cad varied within the grades of ccRCC with no significance between the grades (Figure 5A). On the other hand, significantly lower expression of N-cad was noted as the stages of ccRCC increased, with the normal kidney tissues showing higher expression of N-cad compared to all stages of ccRCCs (Figure 5B). The Kaplan-Meier survival plot and SurvExpress database indicated that the lower expression levels of N-cad were associated with poor OS outcomes in patients (Figure 5C).

The expression of VIM (Figure 4A), an essential marker in ccRCC, was also evaluated in the previously mentioned patient cohorts (in-house and publicly available datasets). The adjacent normal kidney tissue stained with VIM antibody showed strong expression in the glomerulus and proximal tubules (Figure 4A). The primary ccRCC tumour showed moderate positive staining of VIM, while the metastatic ccRCC tumour showed strong positive staining (Figure 4A). Evaluation of the primary ccRCC patient tumours and normal kidney tissues by the CPTAC database showed significantly higher expression of VIM in primary tumours compared to normal kidney tissues (Figure 4B). In addition, in-house primary tumours expressed significantly lower levels of VIM than the metastatic ones (Figure 4C). There was also a stark difference when the matching pairs of primary and metastatic tumours were examined for VIM expression; the matching metastatic tumours had significantly higher expression of VIM than metastatic primary tumours (Figure 4D). In addition, the in-house collected high-grade patient tumours had increased expression of VIM compared to the grade 1 tumour (Figure 5A). Likewise, the higher-stage patient tumours expressed higher VIM levels than their lower-stage counterparts (Figure 5B). Furthermore, the Kaplan-Meier survival plot and GEPIA database demonstrated that the higher expression levels of VIM were significantly associated with the worst OS outcomes in patients (Figure 5C).

Besides the core EMT markers such as E-cad, N-cad, and VIM, we also analysed the expression of other mesenchymal markers, such as TGF-β1 and α-SMA in the above-mentioned ccRCC samples to make the EMT analysis of the above samples stronger.

The IHC expression of TGFβ1 in primary and metastatic RCC tumours and adjacent normal tissue was evaluated by IHC (Figure 6A). The adjacent normal kidney tissue showed few positive patchy expressions in the kidney glomerulus and tubules, while the primary ccRCC tumour showed moderate to strong positive membranous and cytoplasmic staining of TGF-β1. Metastatic RCC tumours showed specific and distinct membranous and cytoplasmic staining of the tumour cells (Figure 6A). The UALCAN database was used to compare the protein expression of the TGF-β1 ligand between the primary ccRCCs and normal kidney tissues (Figure 6B). The expression of TGF-β1 was significantly upregulated in the ccRCC patient’s samples than in the normal adjacent kidney tissues. A significant increase in the expression of TGF-β1 in metastatic ccRCC was observed compared to primaries when explored within the in-house collected patient tumours (Figure 6C). Consistent with that, the matched primary and metastatic ccRCCs, showed an increase in TGF-β1 expression in metastatic tumours compared to primary tumours (Figure 6D). Furthermore, when the association between the TGF-β1 expression was investigated within the tumour grades, it was interesting to note that the grade 1 ccRCC (*n* = 1) showed higher levels of TGF-β1 expression compared to the advanced grades of the ccRCCs (Figure 7A). In terms of stages of tumours, all stages of ccRCC had significantly upregulated TGF-β1 expression compared to normal kidney tissue (Figure 7B). There was no apparent association between the patient’s prognosis with the expression level of TGF-β1 as indicated by the Kaplan Meier plot based on the GEPIA database (Figure 7C).

Figure 6A shows the IHC expression of ACTA-2/α-SMA in primary and metastatic RCC tumours and adjacent normal tissue. The adjacent normal kidney tissue showed a moderate α-SMA expression in the interstitial tissue, while the primary ccRCC tumour showed moderately positive interstitial staining of α-SMA, and metastatic ccRCC tumour displayed a few positive, strong staining of blood vessels in the tumour. Figure 6B shows no difference in the expression of α-SMA in adjacent normal kidney tissues and primary ccRCCs (UALCAN database). Similar expression patterns between the primary and metastatic tumours were observed in in-house collected samples (Figure 6C). However, the matched primary and metastatic tumours showed higher expression of α-SMA than the primary tumours (Figure 6D).

There was no significant difference in the expression of ccRCC tumours (Figure 6D). There was no significant difference in the expression of α-SMA between the tumour grades (Figure 7A) and stages (Figure 7B). Kaplan Meier plotter and the GEPIA database showed no significant association between the level of α-SMA/ACTA-2 expression in the patient’s tumours and OS outcome (Figure 7C).

#### 3.2.3. Other Uncommon EMT Markers

As the aggressiveness and metastatic potential of ccRCC tumours can be predicted based on their biological features, one of them being the expression of EMT markers, we also investigated the signature patterns of various unconventional EMT markers that have been shown to contribute to the EMT process in other cancers. These unconventional markers were evaluated to strengthen the role of EMT in ccRCC progression. These markers were G6PD, ITGAV and AKR1B1.

The IHC expression of G6PD in primary and metastatic RCC tumours and adjacent normal tissue was evaluated (Figure 8A). Adjoining normal kidney tissue showed diffused mild positive staining in the tubules, while ccRCC tumours showed uniform moderate cytoplasmic staining for G6PD (Figure 8A). Metastatic RCC tumour showed moderate membranous and cytoplasmic staining. To explore the relationship between the expression of G6PD and ccRCC, data mining of the CPTAC samples (adjacent normal and ccRCC tumours) available on the UALCAN database was performed. A significant increase in the expression of G6PD was noted in the primary ccRCCs than in the adjacent normal kidney tissues (Figure 7B). There was no significant difference when the expression was compared within the in-house primary and metastatic ccRCC tumours (Figure 8C). However, a pattern of an increased expression of G6PD was noted in matching primary compared to metastatic tumours (Figure 8D). There was a significant increase in G6PD expression in grade 4 compared to grade 2 ccRCC tumours (Figure 9A), while all stages of tumours had significantly higher expression of G6PD than normal kidney tissues (Figure 9B). The overall survival of the ccRCC patients using the GEPIA database and Kaplan Meier plot showed no significant correlation, indicating that the expression of G6PD is not a prognostic marker in ccRCC (Figure 9C).

ITGAV and AKR1B1 have not been studied as EMT promoters in ccRCC. We explored the possibility of the involvement of both these markers in promoting ccRCC progression. In many cancers, such as breast, AKR1B1 has been known to be associated with EMT [39]. A similar link was observed in ccRCC to a certain extent. The IHC expression of AKR1B1 in primary and metastatic RCC tumours and adjacent normal tissue is shown in Figure 8A. The adjacent normal kidney tissue showed moderate AKR1B1 expression in kidney tubules (Figure 8A). Moderate positive staining of AKR1B1 in the primary tumour was also observed (Figure 8A). However, the metastatic RCC tumour showed strong positive staining for AKR1B1 (Figure 8A). Contrary to these observations, extrapolation of ccRCC samples through CPTAC (adjoining normal and ccRCC tumours) demonstrated significantly reduced expression of AKR1B1 in normal kidney tissues compared with its primary ccRCC counterparts (Figure 8B). The in-house primary and metastatic tumours showed an upregulated (not significant) expression pattern of AKRIBI expression in metastatic compared to primary tumours (Figure 8C), which was consistent with AKR1B1 staining in the two matched primary and metastatic tumours (Figure 8D). The expressions of AKR1B1 varied between the grades of ccRCCs (Figure 9A), while significantly high expression of AKR1B1 was observed in different stages of ccRCC compared to normal kidney tissues (Figure 9B). The Kaplan Meier plot deduced from the GEPIA database showed no significant correlation between the survival of the patients and the expression level of AKR1B1 in the patient’s tumours (Figure 9C).

ITGAV, another emerging EMT marker, was also explored for its potential effect on ccRCC progression and prognosis. ITGAV showed an opposite expression profile in comparison with AKR1B1. The IHC expression of ITGAV in primary and metastatic RCC tumours and adjacent normal tissue is shown in Figure 8A. Tumour-adjoining normal kidney tissue showed moderate to strong ITGAV expression in the glomerulus and kidney tubules (Figure 8A). Primary ccRCC tumours showed moderate membranous staining, while no staining was visualized in metastatic tumours (Figure 8A). In the CPTAC samples, no significant difference in the expression pattern of ITGAV was observed between normal kidney tissues and primary ccRCCs (Figure 8B). In the in-house patient tissue cohorts, both the matched and unmatched primary and metastatic pairs, the primary group, showed higher expression of ITGAV (Figure 8C, D). The lower grade 1 (*n* = 1) ccRCC tissue showed a higher expression pattern for ITGAV than the higher grade ccRCCs (Figure 9A). In relation to ccRCC stages, no significant differences in the expression of the ITGAV were observed (Figure 9B). Kaplan-Meir survival curves showed that high expression of ITGAV as a favorable prognostic significance in the OS of ccRCC patients (Figure 9C).

### 3.3. Effect of CSC Markers CD105, CD44 and CD133 in ccRCC Progression and Prognosis

Over recent years, it has been proposed that the stem cell population isolated from the ccRCCs may be important in driving tumorigenesis and resistance to therapy [40]. Traditional cancer treatments like chemotherapy and radiotherapy remove most of the tumour cells from the bulk of the tumour; however, the CSC pool escapes from being eliminated [41,42]. This understanding provides a rationale for targeting these cell populations and might be vital in reversing the chemotherapy and radiation therapy resistance which is an innate nature of ccRCC. Well-defined CSC markers in ccRCC and other cancers may help identify the CSC sub-population and thereby contribute to designing a targeted therapy for ccRCC. This study investigated three crucial established CSC markers, CD44, CD105 and CD133, in ccRCC tumours by in vitro IHC analysis and by using the public domain databases mentioned above. These three markers have proven their importance in other cancers [43,44]. However, the prognostic significance of these markers remains unclear in ccRCC. Studying the level of expression of these CSC markers in different ccRCC patient populations will help identify their underlying role in ccRCC progression and therapy resistance.

The expression of CD44, a transmembrane glycoprotein, was one of the CSCs that was investigated in this study. The IHC expression of CD44 in primary and metastatic RCC tumours and adjacent normal tissue is shown in Figure 10A. Adjacent normal kidney tissue showed negative staining in the glomerulus and moderate cytoplasmic CD44 staining in the tubules, while a distinct membranous and moderate cytoplasmic staining was observed in primary ccRCC (Figure 10A).

On the other hand, metastatic ccRCC showed strong positive staining in membranes and diffused staining of the cytoplasm of tumour cells (Figure 10A). When compared using the CPTAC samples in the UALCAN database, the primary ccRCC patient tissues expressed significantly higher levels of CD44 than the normal kidney tissues (Figure 10B). Next, the in-house patient primary and metastatic tumours were compared for their CD44 expression levels using IHC (Figure 10C). The expression of CD44 was significantly upregulated in metastatic compared to the primary tumours, as demonstrated in the in-house primary and metastatic tumours using IHC (Figure 9C). Consistent with that, the two matching pairs of primary and metastatic tumours, when investigated for their CD44 expression, showed a reduced expression in the matching primary compared with their metastatic counterparts (Figure 10D). The grades and stages of ccRCC tissues were screened for the expression levels of CD44 using in vitro IHC and publicly available CPTAC samples, respectively. The CD44 expression varied in different grades of ccRCC and was insignificant (Figure 11A). The comparison between the normal adjacent kidney tissues and the various stages of ccRCCs showed a consistently significant increase in the expression of CD44 levels in different stages of cancer (Figure 11B). The Kaplan-Meier survival plot and GEPIA database showed that the expression levels of CD44 had no association with the survival outcomes in ccRCC patients (Figure 11C).

The major drawback of the study is the *n* = 1 sample in the grade 1 group of patients, which made analysis for this category of patients challenging. In addition, there was a lack of information regarding the treatment history of patients in the metastatic group. When dealing with human patient samples, the clinical history of patients can be informative in explaining the laboratory research findings. The research outcome can be more robust when the findings can be aligned with essential supporting clinical patient data. Moreover, the treatment of patients can cause some biomarkers in the host to fluctuate. Hence, lack of information on patient treatment in this group of patients can result in an expression profile which would be difficult to align with specific treatments or to the chemo naïve status of the patients [45].

### 3.4. Identification of Prognostic/Diagnostic Signatures

In this study, we analysed a panel of EMT and CSC markers which are known bad prognostic indicators for ccRCC. Even though the expression of some of the markers, such as VIM, TGFβ1, G6PD, AKR1B1, CD44, CD105 and CD133, was significantly high in primary tumours compared to normal kidney tissues, based on CPTAC analysis, the trend did not completely follow in the primary versus metastatic tumour cohort done by in vitro IHC analysis on in-house collected samples. These analyses also showed that expression of VIM, TGFβ1, α-SMA, CD44 and CD133 had significantly high expression in metastatic compared to primary tumours. We also observed significantly low expression of E-cad and N-cad in primary tumours versus normal kidney tissues (CPTAC analysis). Despite these findings, the prognostic significance of these markers, when tested using Kaplan-Meier plots using the GEPIA database, did not provide any prognostic significance except for VIM, which showed bad survival prognosis in patients at high expression, and E-cad, N-cad and ITGAV which presented as bad prognostic indicators when expressed in low levels in ccRCC. The current literature, however, infers contradictory conclusions on these markers and has linked most of these markers with poor prognosis in patients [46,47,48]. To determine if a multi-biomarker signature panel can be developed as an effective screening tool for patients who are at risk of acquiring metastasis and progressive disease, we analysed the combined prognostic significance of a panel of markers that were significantly upregulated in the in-house metastatic versus primary ccRCCs samples. These markers were VIM, TGFβ1, α-SMA, CD44 and CD133. A significant observation to note is that these significantly upregulated markers in the in-house ccRCC metastatic versus primary tumours were found not to have any prognostic significance (except VIM, the high expression had a bad prognosis in patients) when analysed independently by Kaplan Meier plot using the GEPIA database. However, the combined gene expression of these markers showed a significant association of high expression of these markers with poor survival in ccRCC patients in the same dataset (Figure 12A). At the same token, to see if this multi-marker approach would work in developing a diagnostic signature for ccRCC patients, we analysed the combined mRNA expression profile of TGFB1, G6PD, AKR1B1, CD44, CD105, and CD133 using the Kaplan Meier plot and the GEPIA database as these markers showed significant upregulation in primary tumours versus adjoining normal kidney tissues in the CPTAC sample cohort. We demonstrate again that the combined gene expression of these markers showed a significant association with high expression of these markers with poor survival in ccRCC patients even though those markers showed no prognostic significance when analysed independently by the Kaplan Meier plot using the GEPIA database (Figure 12B). These observations infer that a multi-marker rather than a single biomarker approach may be essential to fill the gap caused by the absence of prognostic and diagnostic markers to detect ccRCC patients at risk of metastatic and progressive disease.

## 4. Discussion

Cancer biomarkers constitute a group of biological molecules produced by the tumour-carrying host or the tumours themselves, which signals the initiation and progression of cancer in patients. These molecules can be proteins present in the biological fluids of patients or genomic or proteome changes in the tumours of patients that can be used as diagnostic or prognostic tools to detect the initial diagnosis of the disease, monitor its progression at different stages, and monitor recurrences after treatments. The number of ccRCC cases has been increasing worldwide, and as the numbers grow, the options available for initial diagnosis and monitoring disease status during and after treatment needs further evaluation for precise disease detection and evaluating the specificity and reliability of the new and existing therapeutics. With the discovery of new imaging and genomic technologies, it is now possible to diagnose a greater number of ccRCC patients during routine checkups. Recent genomic characterization of ccRCC has revealed the identification of biologically distinct groups of patients with diverse relapse rates depending on the number of mutations they acquire and has shown that these patients can be stratified to personalized adjuvant treatments and enrolled in new randomized clinical trials [49]. In that context, the primary step in achieving successful treatment outcomes for ccRCC patients would be to understand the proteome as well as genome of ccRCC tumours using multiple candidates and omics-based combinational approaches and subsequent validation of that analysis on a comprehensive cohort of patients with diverse demographics at the publicly available platform, such as CPTAC, TCGA, etc., which contains complete molecular information of ccRCC. This may result in developing innovative descriptors/signatures which can have a wide application as diagnostic and prognostic biomarkers that ultimately can lead to targeted treatment options in the realm of ccRCC. The current study thus focused on the biomarkers of important physiological processes such as EMT and CSC that play crucial roles in the pathogenesis of ccRCC, with the aim to develop diagnostic and prognostic markers for ccRCC patients, a subtype of cancer responsible for >85% of renal cancer patients, with high 5-year morbidity (median survival of about 13 months and 5-year survival under 10%), resulting in significant challenges in patient care.

The initial oncoprint representation of ccRCC patients showed that these patients had different types of mutations in the selected EMT and CSC-associated genes we queried in this study. Hypoxia is an important environmental factor responsible for promoting tumour aggressiveness and metastasis through enhanced expression/activation of HIF-related genes [50]. Mutational inactivation of the tumour suppressor VHL gene is an early genetic anomaly in the mainstream ccRCCs, leading to enhanced expression of the HIF-1α and HIF-2α transcription factors. Expression analysis of HIF-1α and HIF-2α on human ccRCCs and functional studies on human ccRCC cell lines have suggested HIF-1α as a suppressor and HIF-2α as a facilitator of invasive tumour behaviour [51]. However, their roles as diagnostic and prognostic indicators have not been functionally addressed. We show that normal kidney tubules contain weak to strong staining of both HIF-1α and HIF-2α, which was retained in metastatic tumours, different grades, and stages of tumours. Contrary to that, a good prognosis in the context of OS was obtained only using the TCGA dataset with high expression of HIF-2α in ccRCC patients. This observation is contradictory to the described angiogenic role of HIF-2α, which occurs through the upregulation of VEGF expression [52]. In that context, a recent ccRCC model described HIF-1α to be essential for tumour initiation and proliferation by regulating glucose uptake and glycolysis, whereas HIF-2α deletion had only minor effects on tumour development and spread but regulated genes associated with lipoprotein metabolism, ribosome biogenesis and E2F and MYC transcriptional activities [12]. HIF-2α-deficient tumours in that study were also characterised by enhanced antigen presentation, interferon signalling and CD8+ T cell infiltration and activation, suggesting an immune cell evading role of HIF-2α in ccRCC progression. This may reflect the expression pattern of HIF-1α and HIF-2α staining we show, suggesting a role of both HIF-1α and HIF-2α in ccRCC progression and disease persistence [12].

EMT is the key process that facilitates local tumour invasion and distant migration of cancer cells leading to metastasis, acquisition of drug resistance, host immune escape and maintenance of cancer stemness [53]. In our study, the conventional hypothesis of EMT reliant on the pattern of repression of E-cad expression along with enhanced expression of VIM holds true for ccRCC. In the current study, although not significant, both the primary and metastatic tumours across all the datasets do show conventional glimpses of EMT in relation to E-cad and VIM expression. This classical EMT pattern is significantly observed between the adjacent normal kidney tissues versus primary tumours. These observations are consistent with previous studies in ccRCC, which reported E-cad and VIM as independent prognostic factors for progression-free and overall survival in ccRCC patients [9]. However, the high expression of E-cad in the primary compared to the metastatic tumours contradicts other previous studies, which reported significantly increased E-cad expression in metastatic breast cancer in the bone compared to primary tumours [54], and an increased expression of E-cad in metastatic lung nodules compared to primaries [55]. In that setting, even though a loss of E-cad is the first step towards achieving EMT, downregulation of E-cad expression in many circumstances is inadequate to initiate EMT in tumour cells [56]. These inconsistencies may exist due to epithelial-mesenchymal plasticity (EMP) in cancer cells [57], by which the tumour cells have the flexibility in moulding towards either the epithelial or mesenchymal or a hybrid state, depending on the stimulus they receive from the tumour microenvironment [58]. These hybrid EMT-like cells stuck in the transitory state are highly plastic, aggressive, ankiois and chemotherapy-resistant and possess stem cell-like characteristics [57,59,60]. Hence, an EMT phenomenon is not a dualistic process involving the complete transformation of cells from an epithelial to a mesenchymal state, rather is a gradual cellular process in which, in each cellular state, cancer cells express different levels of epithelial and mesenchymal attributes and exhibit a different spectrum of morphological, transcriptional, and epigenetic features which are between the two distinct poles of epithelial and mesenchymal parameters [61,62,63]. Hence, it is increasingly important to understand these hybrid EMT-like cells from the biomarker and targeted therapeutic perspectives.

Previous studies have shown that enhanced expression of mesenchymal markers, like N-cad and VIM, are important proteins in initiating EMT in tumours like the bladder, breast, colon, and prostate [64,65,66,67]. The upregulated expression of N-cad has been associated with increased invasiveness and motility in cancer cells [68,69]. In the current study, the enhancement of N-cad expression in metastatic RCCs compared to primary tumours in in-house collected samples may indicate that an upregulated N-cad expression may be required for metastasis in ccRCC. This is consistent with the upregulated expression of N-cad and the association of increased migratory and invasive abilities of cancer cells [69,70]. The current literature on ccRCC shows very few studies in which the expression of N-cad has been studied. Consistent with our data, one recent study has shown high N-cad expression to be associated with significantly larger tumour size, higher nuclear grade, and tumour necrosis in a cohort of ccRCC patients using tissue array [71]. However, these data are contrary to another previous study that showed a normal pattern of N-cad expression despite the elevated tumour grades, and patients with N-cad normal expression have a poorer prognosis than those with N-cad abnormal expression, suggesting that N-cad may play a different role in ccRCC and may not be directly associated with the malignant potential of ccRCC [72].

To get a more precise representation of the existence of EMT and in search for new prognostic markers, we investigated the expression of other unconventional EMT markers not previously investigated in ccRCC from the EMT point of view. This study investigated TGFβ1, α -SMA, ITGAV, AKR1B1 and G6PD as potential EMT markers in ccRCC. Consistent, significant, and gradual elevation in the expression of TGFβ1 in primary tumours versus normal kidney tissues; and between metastatic and primary tumours, and in different stages of ccRCC compared to normal tissues indicate a role of TGFβ1 in the progression of ccRCC. This expression profile of TGFβ1 is consistent with the defined role of TGFβ1 in the initiation of EMT-related progression in other cancers [15,73]. On the other hand, even though the expression of α-SMA contributes to EMT in normal epithelial cells during embryogenesis and wound healing [74], and also in certain cancers [20], no change in the expression of α-SMA between primary tumours versus normal kidney tissues; and between metastatic and primary tumours; and in different stages of ccRCC indicates that α-SMA may not have an active role in EMT-induced cancer progression in ccRCC.

Enhanced expression of certain subsets of integrins has been shown to be involved with matrix remodelling required for the EMT transformation of cancer cells [75]. In the current study, ITGAV, which has been involved in increased invasion, proliferation, and self-renewal in many cancers [76], was investigated for its EMT potential in ccRCC. The fact that there was no ITGAV expression difference between primary tumours and normal kidney tissues and the ITGAV levels were highest in the lower grade than the higher-grade tumours; high expression of ITGAV has a favourable prognostic significance in ccRCC patients infers ITGAV to be a favourable prognostic marker. However, ITGAV has been linked to shorter OS in patients with oesophagal adenocarcinoma and breast cancer [76,77]. Recent studies also suggest that ITGAV is involved with TGFβ1 activation [18]. In addition, ITGAV potentiates the capabilities of TGFβ1 to down-regulate E-cad expression in RCC [78]. However, in this study, no overall correlation between ITGAV and TGFβ1 could be derived as the expression of TGFβ1 followed a classical tumour progressive trend (was higher in primary tumours versus normal kidney tissues and was also high in metastatic versus primary tumours. In addition, TGFβ1 expression progressively enhanced with stages of RCC, suggesting a role of TGFβ1 in the progression and spread of the disease). An interesting observation was made in grade 1 ccRCCs (*n* = 1) collected in-house. This patient had increased ITGAV and TGFβ1 levels that negatively correlated with the E-cad expression. These observations may hint at crosstalk between TGFβ1, ITGAV and E-cad, which can be grade-dependent in ccRCC; however, this warrants further investigation.

AKR1B1, an enzyme involved in the reduction of the aldehydic group of lipid peroxidation end-product, has been implicated in many cancers and has been linked to processes like inflammation, EMT and cancer [16]. Although the expression of AKR1B1 was elevated in primary ccRCC versus normal kidney tissues and positively correlated with the high stages of tumours versus normal tissue, it was not prognostically significant. Another metabolic marker implicated with EMT in cancers is G6PD [79]. G6PD is a rate-limiting enzyme in the pentose phosphate pathway (PPP) and plays a key role in maintaining the redox potential in cells by converting NADP+ to NADPH. G6PD has been associated with increased tumour cell migration, proliferation, invasion, and colony formation capacity, all of which are defined characteristics that lead to EMT in cancer cells [79]. Even though the expression of G6PD was significantly high in primary tumours compared to normal kidney tissues, and a similar trend was noted in normal tissues versus different stages of ccRCC, it was not a good prognostic indicator in the context of OS in ccRCC patients. Contrary to that, overexpression of G6PD represented a potential prognostic factor in ccRCC in a recent study [80]. The same research group showed that G6PD is expressed higher in the advanced stages and grades of ccRCC [81]. These contradictory findings in the expression of G6PD in ccRCC need further evaluation.

EMT exists in the cancer stem cell population, which allows this population of cells to escape cancer treatment resulting in drug resistance and metastasis [23]. ccRCC tumours expressing EMT markers have been shown to display CSC characteristics [82]. The status of mesenchymal stem cell markers, CD105 and CD44, were significantly elevated in expression in the metastatic ccRCC compared to primary tumours. While the expression of CD105 positively correlated with the higher grades of ccRCC, no such association was observed in the case of CD44 expression. In contrast, in the case of ccRCC stages, the increasing stages correlated with increased levels of CD44 expression, but then varied levels of expression for CD105 were observed. Previous studies have shown that upregulated CD105 is associated with higher tumour stages [83]. On the other hand, other studies have concurred that CD105 is an unfavourable prognostic marker in ccRCC and other cancers [84,85]. The current study showed that CD105 has a positive prognostic potential while CD44 is non-prognostic in ccRCC patients. CD133, on the other hand, is a hematopoietic stem cell marker and was significantly elevated in the metastatic ccRCC tumours. CD133 has been associated with tumour vascularisation, and as ccRCC has vascular characteristics, it can be a useful angiogenic marker in ccRCC. The overexpression of CD133 was associated with longer survival outcomes in this study which concurs with previous studies [48].

## 5. Conclusions

RCC (>85% of which is ccRCC) constitutes 2% of global cancers, is one of the 10 most common cancers in the United States and is the fastest-growing cancer in North America [1]. Over the last few years, our knowledge of the mutational changes in ccRCC has broadened, but most of these genomic changes are not easily targetable by drugs and do not capture the early or late-stage features of the disease, making it hard to treat patients.

In this study, we show the development of multi-marker diagnostic and prognostic signatures for ccRCC patients. We have been able to show that combination of molecular characteristics obtained at the protein and mRNA levels from large-scale public domain datasets and from in vitro analysis of a small cohort of human tumours accrued in-house from different ccRCC patients have the potential to develop distinct EMT/CSC associated signature patterns essential to identify patients in high-risk cases for early diagnosis so that appropriate treatment can be provided. Considering most of the ccRCC patients, after diagnosis, under current clinical practice, are clinically observed for tumour growth for prolonged periods without any intervention, screening this patient group for determination of these signatures in tumours immediately after diagnosis may result in identifying high-risk patients. This would result in a paradigm shift in the clinical management of ccRCC patients and would increase the progression-free and enhanced quality of life in high-risk patients, with subsequent lower physical, emotional, and economic burdens on society. With the recent advent of EMT inhibitors in clinical trials [86], the above option of screening may be an effective way for future patient care. Figure 13 shows the graphical representation of the method for developing the multi-marker-based approach discussed in this study.

## Figures and Tables

**Figure 1 cancers-15-02586-f001:**
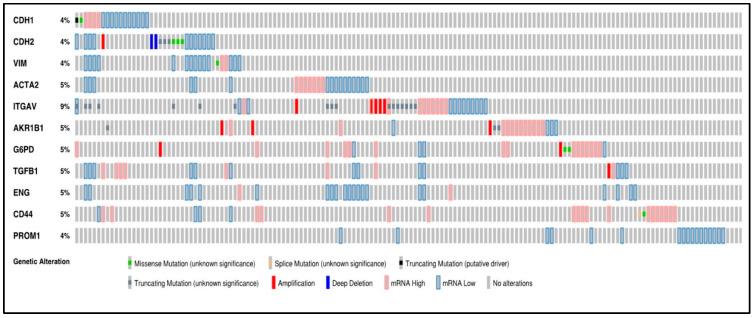
Oncoprint analysis of *n* = 538 ccRCC patients, showing alterations in the gene expression of 11 marker genes associated with EMT and CSC. Rows and columns depict genes and ccRCC patients, respectively. Genomic alterations such as copy number alterations (deep deletions and amplification), mutations such as missense, splice and truncating mutations, and up and down-regulation of mRNA are summarised by glyphs and coding. The cases are represented according to alterations. The analysis provides a summary of genomic alterations (legend) (rows) affecting individual patients (columns). The mutational frequency is labelled on the left in percentage.

**Figure 2 cancers-15-02586-f002:**
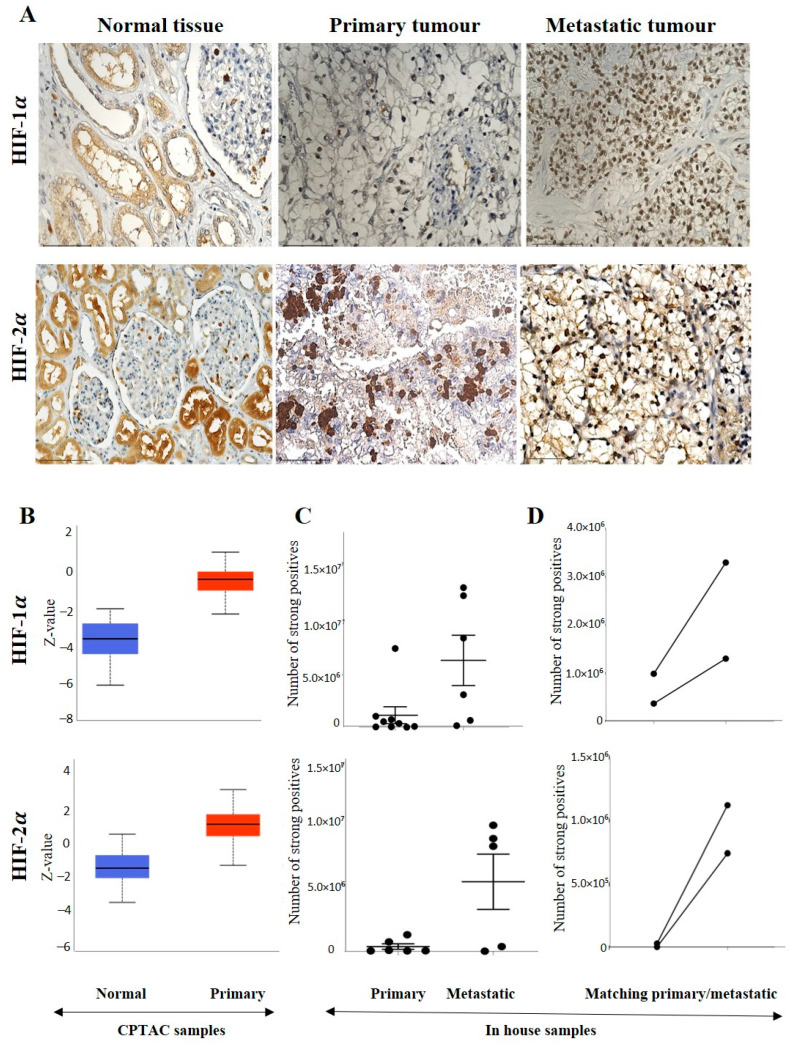
(**A**) IHC representation of HIF-1α and HIF-2α proteins in normal kidney tissue, primary and metastatic tumours. Magnification: 40× Scale bar: 50 μm. (**B**) In silico investigation of HIF-1α and HIF-2α protein expression levels between adjacent normal kidney (*n* = 84) vs. primary ccRCC (*n* = 110) in CPTAC samples. Z-values represent standard deviations (SD) from the median across the RCC samples. Log2 spectral count ratio values from CPTAC were first normalised within each sample profile and then normalised across samples. Statistical method: Z-test; no significance was observed. (**C**) Expression of HIF-1α and HIF-2α in in-house collected ccRCCs; primary (*n* = 11) and metastatic (*n* = 5–6) tissues by IHC; no significance, student’s t-test was used. (**D**) Correlation of HIF-1α and HIF-2α expression on matching primary (*n* = 2, grade 2) and metastatic ccRCC (*n* = 2, grade 3) in in-house accrued samples.

**Figure 3 cancers-15-02586-f003:**
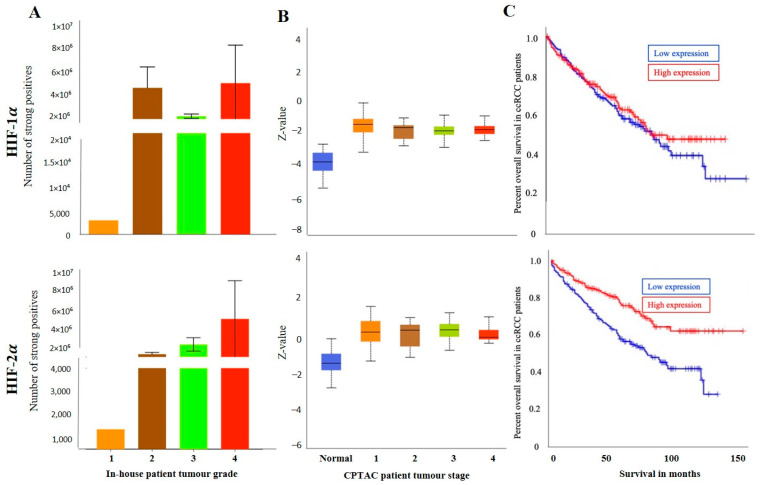
(**A**) HIF-1α and HIF-2α expression levels by IHC, according to grades, in in-house collected ccRCCs [grades-1 (*n* = 1), 2 (*n* = 6), 3 (*n* = 9), 4 (*n* = 3)], no significance was found by student’s *t*-test between the groups. (**B**) HIF-1α and HIF-2α expression levels, according to stages [1 (*n* = 52), 2 (*n* = 13), 3 (*n* = 33), 4 (*n* = 12)] of cancer in CPTAC ccRCC samples. Significance was tested by student’s *t*-test between two groups at a time. No significance was achieved. (**C**) Kaplan-Meier survival curves in patients expressing high [red] (*n* = 258) and low [blue] (*n* = 258) levels of HIF-1α and HIF-2α expression; *p* < 0.001 between low and high HIF-2α expression.

**Figure 4 cancers-15-02586-f004:**
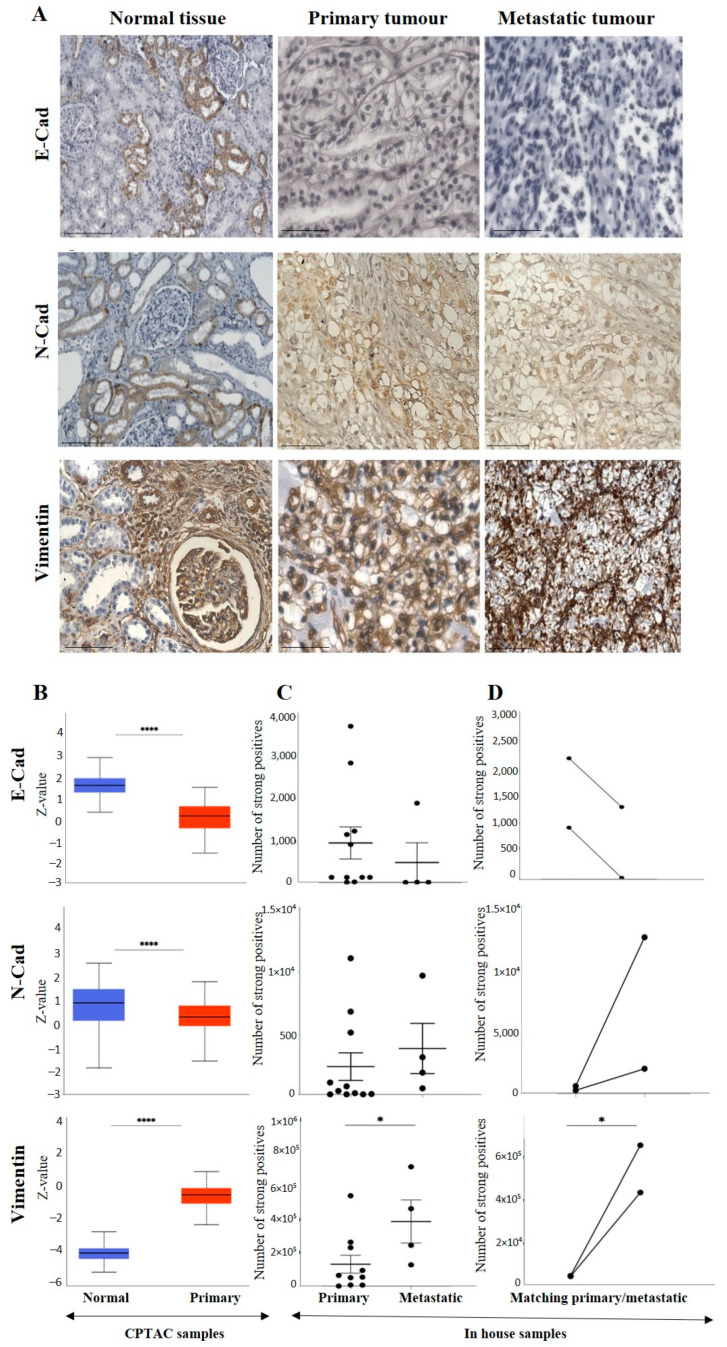
(**A**): Representative IHC demonstration of the epithelial marker, E-cad, mesenchymal markers N-cad and VIM in adjacent normal kidney tissue, primary and metastatic RCC tumours stained with respective E-cad, N-cad and VIM primary antibodies. Magnification: 40×, Scale bar: 50 μm. (**B**) In silico investigation of E-cad, N-cad and VIM expression between adjacent normal kidney (*n* = 84) vs. primary ccRCC tumours (*n* = 110) in CPTAC samples. Z-values represent standard deviations (SD) from the median across the RCC samples. Log2 spectral count ratio values from CPTAC were first normalised within each sample profile and then normalised across samples. Statistical method: Z-test; **** *p* < 0.0001. (**C**) Expression of E-cad, N-cad and VIM in in-house collected ccRCC primary (*n* = 10–11) and metastatic (*n* = 4) tumours by IHC. No statistical significance was obtained by Student’s *t*-test, except for VIM, * *p* < 0.05. (**D**) Correlation of E-cad, N-cad and VIM expression on matching primary (*n* = 2, grade 2) and metastatic ccRCC (*n* = 2, grade 3) in-house accrued samples.

**Figure 5 cancers-15-02586-f005:**
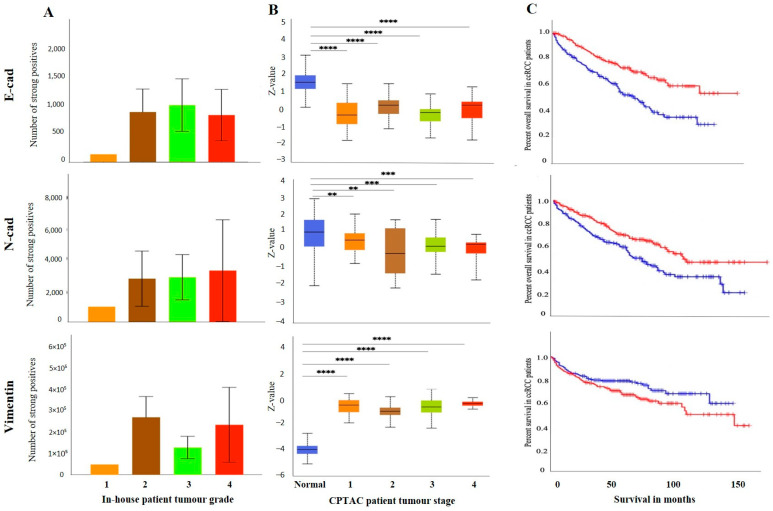
(**A**) E-cad, N-cad and VIM expression levels by IHC, grade-wise, in in-house collected ccRCCs, no significance found by Student’s *t*-test between the groups. (**B**) E-cad, N-cad, and VIM expression levels, stage-wise in CPTAC ccRCC samples. Significance was tested by Student’s *t*-test, normal kidney tissue vs. stage 1, stage 2, 3 and 4 tumours, ** *p* < 0.01, *** *p* < 0.001, **** *p* < 0.0001. Sample numbers in each grade and stage in (**A**,**B**) are described in Figure 3A,B. (**C**) Kaplan-Meier survival curves in patients expressing high [red] (*n* = 258) and low [blue] (*n* = 258) levels of E-cad, N-cad, and VIM; *p* < 0.05, *p* < 0.001 between high and low expression.

**Figure 6 cancers-15-02586-f006:**
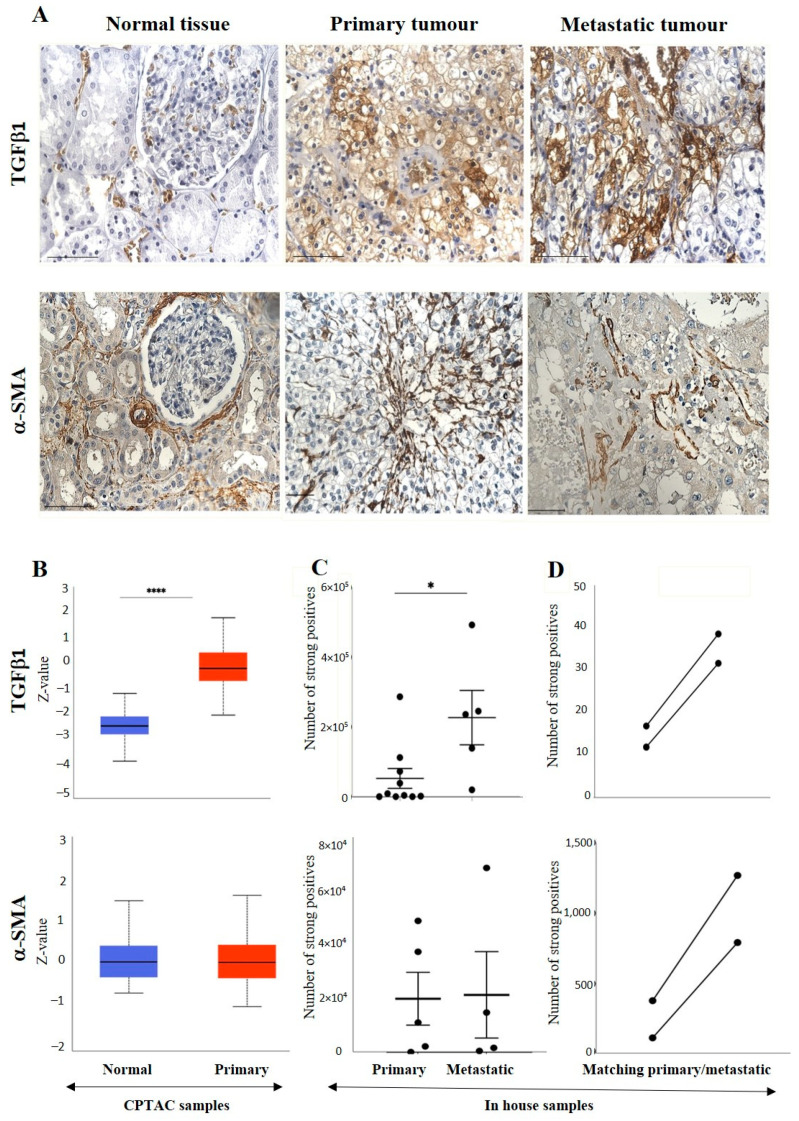
(**A**) IHC representation of TGFβ1 and α-SMA in adjacent normal kidney tissue, primary and metastatic RCC tumours stained with TGF-β1 and α-SMA specific primary antibodies. Magnification: 40×, Scale bar: 50 μm. (**B**) In silico investigation of TGFβ1 and α-SMA expression between adjacent normal kidney (*n* = 84) vs. primary ccRCC tumours (*n* = 110) in CPTAC samples. Z-values represent standard deviations (SD) from the median across the RCC samples. Log2 spectral count ratio values from CPTAC were first normalised within each sample profile and then normalised across samples. Statistical method: Z-test; significance is indicated by **** *p* < 0.0001 for TGFβ1 expression between normal kidney tissues and primary ccRCCs. (**C**) Expression of TGFβ1 and α-SMA in in-house collected ccRCC primary (*n* = 5) and metastatic (*n* = 4) tumours by IHC. Statistical significance was obtained with TGF-β1 expression between primary and metastatic tumours using student’s *t*-test; * *p* < 0.05. (**D**) Correlation of TGFβ1 and α-SMA expression on matching primary (*n* = 2, grade 2) and metastatic ccRCC (*n* = 2, grade 3) in-house accrued samples.

**Figure 7 cancers-15-02586-f007:**
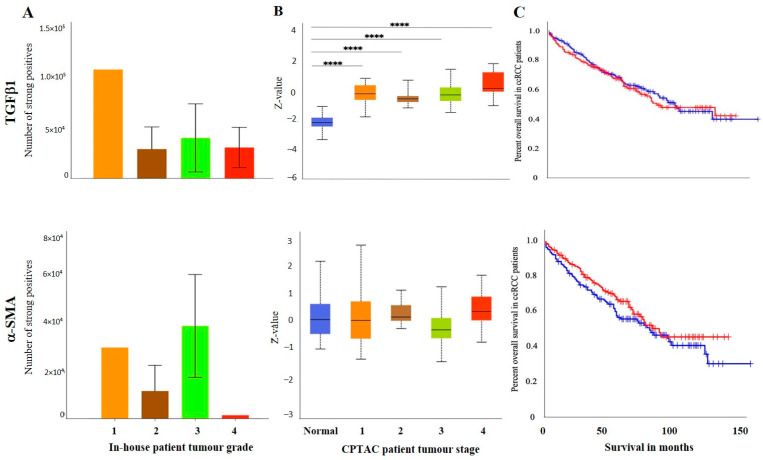
(**A**) IHC expression levels, grade-wise, of TGFβ1 and α-SMA in in-house collected ccRCCs. No significance was found by Student’s *t*-test between the groups. (**B**) TGF-β1 and α-SMA expression levels, stage-wise in CPTAC ccRCC samples. Significance was tested by student’s t-test, normal kidney tissue vs. stage 1, stage 2, 3 and 4 tumours: **** *p* < 0.0001 for TGFβ1. The sample number in each grade and stage in (**A**,**B**) are as in Figure 3A,B. (**C**) Kaplan-Meier survival curves in patients expressing high [red] (*n* = 258) and low [blue] (*n* = 258) levels of TGF-β1 and α-SMA. No significance was determined.

**Figure 8 cancers-15-02586-f008:**
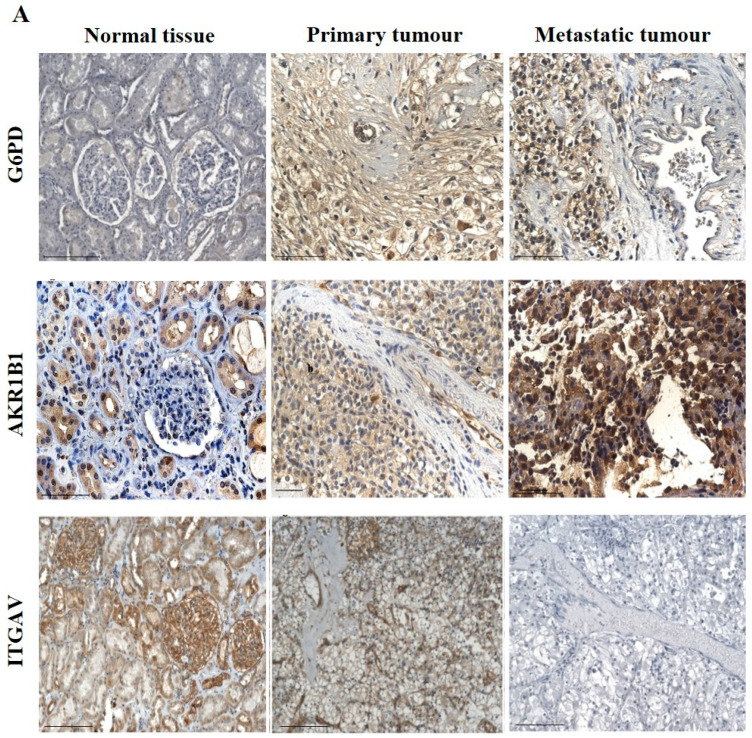
(**A**) IHC representation of G6PD, AKR1B1 and ITGAV in adjacent normal kidney tissue, primary and metastatic RCC tumours stained with the specific primary antibodies. Magnification: 40×, Scale bar: 50 μm. (**B**) In silico investigation of G6PD, AKR1B1 and ITGAV expression between adjacent normal kidney (*n* = 84) vs. primary ccRCC tumours (*n* = 110) in CPTAC samples. Z-values represent standard deviations (SD) from the median across the RCC samples. Log2 spectral count ratio values from CPTAC were first normalised within each sample profile and then normalised across samples. Statistical method: Z-test; **** *p* < 0.0001 for G6PD and AKR1B1 expression between normal kidney tissues and primary ccRCCs. (**C**) Expression of G6PD, AKR1B1 and ITGAV in in-house collected ccRCC primary (*n* = 7) and metastatic (*n* = 4) tumours by IHC. No statistical significance was obtained by using Student’s *t*-test. (**D**) Correlation of G6PD, AKR1B1 and ITGAV expression on matching primary (*n* = 2, grade 2) and metastatic ccRCC (*n* = 2, grade 3) in-house accrued samples.

**Figure 9 cancers-15-02586-f009:**
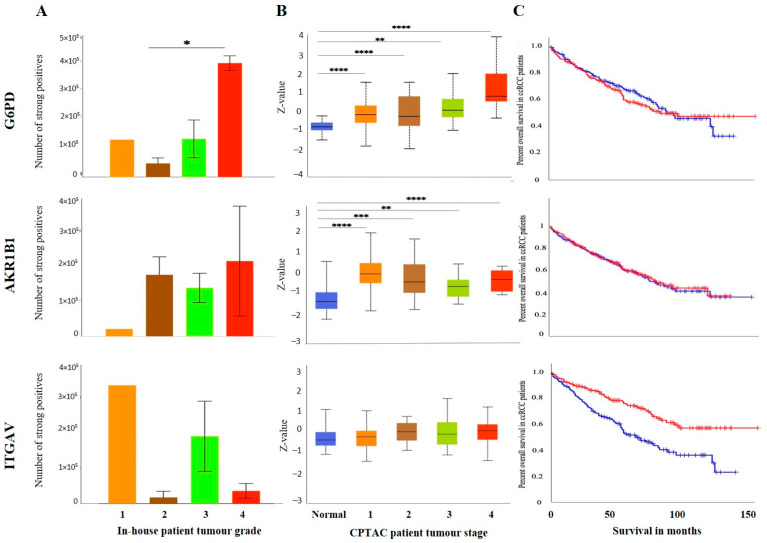
(**A**) IHC expression levels, grade-wise, of in G6PD, AKR1B1 and ITGAV in-house collected ccRCCs. Statistical significance in G6PD expression was found by Student’s *t*-test between grades 2 and 4 patients, * *p* < 0.05. (**B**) G6PD, AKR1B1 and ITGAV expression levels, stage-wise, in CPTAC ccRCC samples. Significance was tested by Student’s *t*-test, normal kidney tissue vs. stage 1, stage 2, 3 and 4 tumours: ** *p* < 0.01,*** *p* < 0.001 and **** *p*<0.0001 for G6PD and AKR1B1. Sample numbers in each grade and stage in (**A**,**B**) are described in Figure 3A,B. (**C**) Kaplan-Meier survival curves in patients expressing high [red] (*n* = 258) and low [blue] (*n* = 258) levels of G6PD, AKR1B1 and ITGAV. Significance was determined for ITGAV, *** *p* < 0.001.

**Figure 10 cancers-15-02586-f010:**
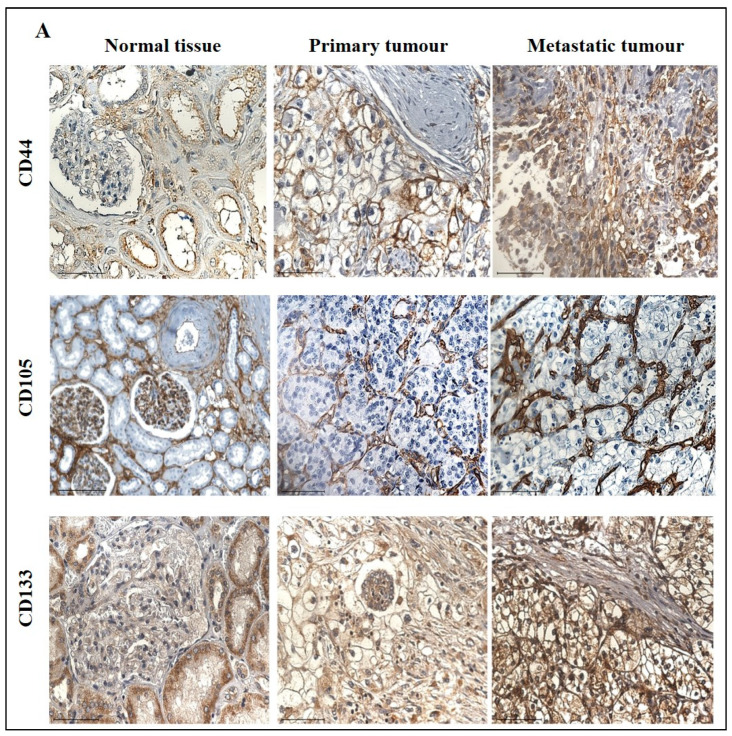
(**A**) Representative IHC demonstration of the CSC marker CD44, CD105 and CSC133 in adjacent normal kidney tissue, primary and metastatic RCC tumours stained with respective primary antibodies. Magnification: 40×, Scale bar: 50 μm. (**B**) In silico analyses of CD44, CD105 and CD133 expression between adjacent normal kidney (*n* = 84) vs. primary ccRCC tumours (*n* = 110) in CPTAC samples. Z-values represent standard deviations (SD) from the median across the RCC samples. Log2 spectral count ratio values from CPTAC were first normalised within each sample profile and then normalised across samples. Statistical method: Z-test; * *p* < 0.05, **** *p* < 0.0001. (**C**) Expression of CD44, CD105 and CD133 in in-house collected ccRCC primary (*n* = 11) and metastatic (*n* = 4) tumours by IHC. Statistical significance was obtained by Student’s *t*-test, * *p* < 0.05. (**D**) Expression of CD44, CD105 and CD133 on matching primary (*n* = 2, grade 2) and metastatic ccRCC (*n* = 2, grade 3) in-house accrued samples.

**Figure 11 cancers-15-02586-f011:**
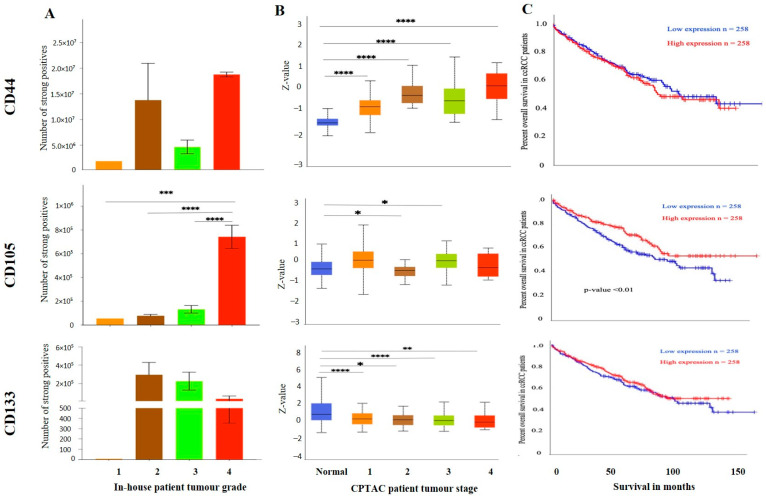
(**A**) IHC expression levels, grade-wise, of CD44, CD105 and CD133 in-house collected ccRCCs. Statistical significance in CD105 expression was found between grade 2, grade 3 and grade 4 samples by Student’s *t*-test, *** *p* < 0.001, **** *p* < 0.0001. (**B**) CD44, CD105 and CD133 expression levels, stage-wise, in CPTAC ccRCC samples. Significance was tested by Student’s *t*-test, normal kidney tissue vs. stage 1, stage 2, 3 and 4 tumours: * *p* < 0.05, ** *p* < 0.01 and *** *p* < 0.001. Sample number in each grade and stage in (**A**,**B**) are as in Figure 3A,B (**C**) Kaplan-Meier survival curves in patients expressing high [red] (*n* = 258) and low [blue] (*n* = 258) levels of G6PD, AKR1B1 and ITGAV. No significance was determined.

**Figure 12 cancers-15-02586-f012:**
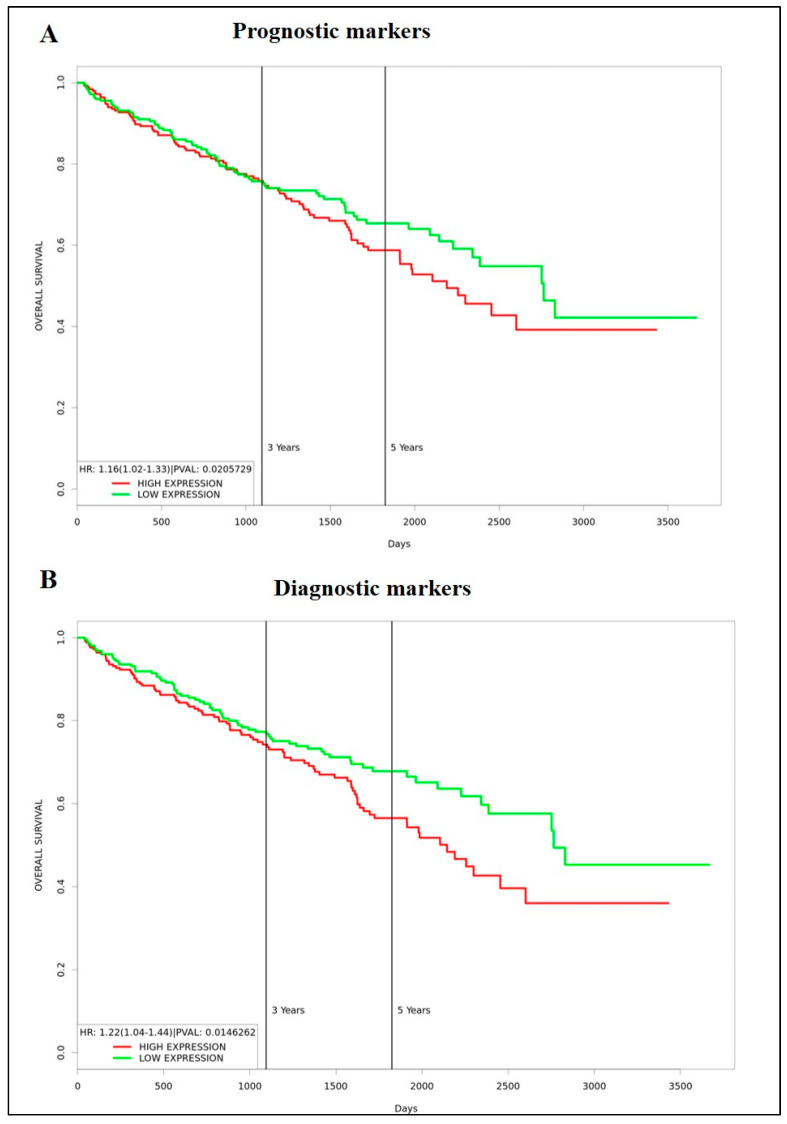
(**A**) Kaplan-Meier survival plot generated using the TCGA KIRC dataset and PROGgeneV2 web-based tool. *n* = 253 (low expression, green) *n* = 253 (high expression, red), *p* < 0.05. Combined expression of VIM, α-SMA, TGFB1, CD44, PROM1. (**B**) Kaplan-Meier survival plot generated using the TCGA KIRC dataset and PROGgeneV2 web-based tool. *n* = 253 (low expression, green) *n* = 253 (high expression, red), *p* < 0.05. Combined expression of TGFB1, G6PD, AKR1B1, CD44, ENG (CD105) and PROM1. Cohort divided by the median of gene expression.

**Figure 13 cancers-15-02586-f013:**
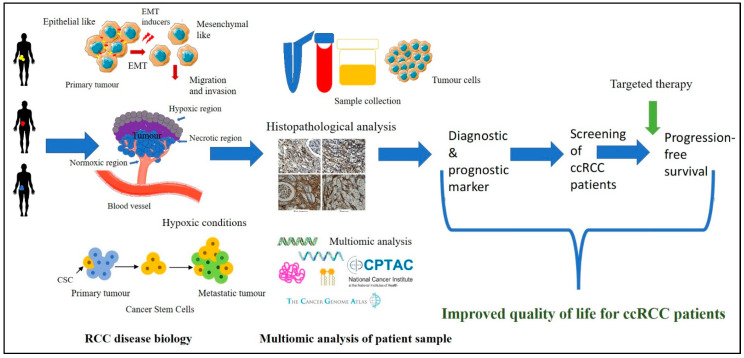
ccRCC disease biology is interlinked with the biological processes of hypoxia-associated EMT and CSC. Access to the patient’s tumours and their analysis by immunohistochemistry for hypoxia-associated EMT/CSC markers, in combination with the data available in public domain platforms such as CPTAC and TCGA, can enable us to get insights into the complex tumour landscape. The knowledge gained using these multi-disciplinary approaches can be initiated for the development of a multi-marker diagnostic/prognostic panel which can be used to screen ccRCC patients at an early stage leading to the stratification of high-risk patients with the potential of rapid disease progression. This screening approach, in combination with targeted combination therapy, may result in better treatment outcomes for ccRCC patients.

**Table 1 cancers-15-02586-t001:** Description of ccRCC patients participating in immunohistochemistry analysis.

Patient Number	Gender	Age at Diagnosis (Years)	Type of Tumour	Kidney Involved	Tumour Stage (TNM Classification)	Tumour Grade	Metastatic Tumour Site If Present
1	M	74	Primary	Right	pT2a NX	3	NA
2	F	78	Metastatic	Right (primary)	UNK	3	Liver
3	M	69	Primary	Left	pT3aNXMX	3	NA
4	M	75	Primary	Left	pT3aNXMX	1	NA
5	F	52	Primary	Right	pT1b NX MX	2	NA
6	M	71	Primary	Right (primary)	UNK	4	Lung
7	M	65	Primary	Left	pT3a NX	3	NA
8	F	51	Primary	Right	pT1a NX	4	NA
9	F	UNK	Primary	Left	UNK	3	NA
10	F	UNK	Metastatic	Left (primary)	UNK	3	Adrenal gland
11	F	UNK	Primary	Left	pT3aNX	3	NA
12	M	UNK	Metastatic	Right (primary)	T3aNXMX	2	Pancreas
13	M	65	Metastatic	UNK	UNK	3	Lung
14	F	48	Metastatic	UNK	UNK	3	Lung
15	UNK	UNK	Primary	UNK	UNK	3	UNK
16	UNK	UNK	Primary	UNK	UNK	2	UNK
17	UNK	UNK	Primary	UNK	UNK	4	UNK
18	UNK	UNK	Primary	UNK	UNK	2	UNK
19	UNK	UNK	Metastatic	UNK	UNK	2	UNK

UNK: Unknown. NA: Not applicable. MX: Metastasis cannot be measured. NX: There is no information about the nearby lymph nodes. pT1a: Primary tumour subtype a, stage 1 tumour ≤ 4cm and has not metastasized. pT1b: Primary tumour subtype a, stage 1 tumour between 4–7 cm. pT2a: Primary tumour stage 2: >7 cm and ≤10 cm. pT3a: invades renal vein/branches, perineal fat, renal sinus fat or pelvicalyceal system.

**Table 2 cancers-15-02586-t002:** Gene and protein nomenclature of the markers studied.

Sr No.	Markers	
	Genes	Proteins
1	*CDH*1	E-cadherin [13,14]
2	*CDH*2	N-cadherin [13,14]
3	*Vim*	Vimentin [13,14]
4	*ACTA*2	α-SMA [20]
5	*TGFB*1	TGFβ1 [11,15]
6	*G6PD*	G6PD [19]
7	*AKR1B*1	AKR1B1 [16,19]
8	ITGαv	ITGAV [17,18,19]
9	*CD*44	CD44 [27,28]
10	*ENG*	CD105 [27,28]
11	*PROM*1	CD133 [28,29]

**Table 3 cancers-15-02586-t003:** List of in vitro and in silico tools used in this study.

Number	Expression of Tissues	Tools Used	Dataset Used	Protein/mRNA Expression Analysed
1	Oncoprint analysis	cBioPortal	TCGA Firehose legacy	Genes and mRNA
2	Normal vs. primary tissues	UALCAN	CPTAC (clear cell RCC)	Protein
3	Primary vs. metastatic tissues	IHC	In-house (non-matching ccRCC tissues)	Protein
4	Primary vs. metastatic tissues	IHC	In-house (matching ccRCC tissues)	Protein
5	Grades of ccRCC	IHC	In-house ccRCC tissues	Protein
6	Stages of ccRCC	UALCAN	CPTAC (clear cell RCC)	Protein
7	Kaplan Meier survival curves of single genes	GEPIA	TCGA (KIRC-ccRCC dataset)	mRNA
8	Kaplan-Meier survival curve—Combination gene panel	PROGgeneV2	TCGA (KIRC-ccRCC dataset)	mRNA

## Data Availability

Most of the data used in this manuscript are available in the public domain. The name of the datasets is indicated in the manuscript, and access to these data can be gained if required. The in-house generated data by the authors can be obtained with a reasonable request.

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
