# Peer review of "In Vitro and In Silico Analysis of Epithelial-Mesenchymal Transition and Cancer Stemness as Prognostic Markers of Clear Cell Renal Cell Carcinoma"

_cancers, 2023, doi:10.3390/cancers15092586_

Round 1

Reviewer 1 Report

Clear cell renal cell carcinoma (ccRCC) is a highly aggressive cancer responsible 14 for about 85% of kidney cancers. In the current study the authors used data from in-house collected patient’s tumours and public domain datasets to identify EMT and CSC markers that may be to be prominent players in ccRCC progression. Using these analysis the authors show the development of multi-marker diagnostic and prognostic signatures which may stratify high risk patients likely to progress to a metastatic disease.They investigated the genetic alteration of a panel of 11 selected genes in ccRCC patient tumours using oncoprint analyses from cBioPortal. No prognostic significance of HIF-1a expression but significance of HIF-2a expression was noted on the OS of the patients. Low expression levels of E-cad and N-cad were significantly associated with poor OS. But higher expression levels of VIM was significantly associated with the worst OS outcomes in patients. Kaplan Meier plotter and the  GEPIA database showed no significant association between the level of α-SMA/ACTA-2 expression in patient’s tumours and OS outcome. Among EMT markers G6PD, ITGAV and AKR1B1, Kaplan-Meir survival curves showed that high expression of ITGAV as a favorable prognostic significance in the OS of ccRCC patients. For the CSC markers CSC markers CD44, CD105 and CD133 there was no association with the survival outcomes in ccRCC patients. The authors conclude that a multi-marker approach may be essential to fill the gap caused by the absence of prognostic and diagnostic markers to detect ccRCC patients at a risk of metastatic and progressive disease.

Can the authors elaborate on the selection of the markers they used for the study?

Author Response

We thank the reviewer for his comment.  Elaboration on the markers has been done throughout the manuscript.  Please see lines 59-70, lines 97-108, lines 109-134 in the Introduction of the manuscript.   In addition, in the revised manuscript we have now further re-introduced the markers in the final paragraph of the Introduction for more emphasis.  Further to that, in page 6, Table 2 we have provided references with each marker to help the readers with the background on each marker. On top of that, there are additional references in the Discussion section on the background of each marker, of its role in EMT/CSC and cancer progression. 

Reviewer 2 Report

Comment to the Authors:  

The authors should be congratulated for their work. It explores patients' perspective and needs in detail.

Of course, the more patients are included in such studies, the stronger results are and greater impact they can have on clinical practice and patient management. This work is very sophisticated and takes in consideration a great number of biological factors that may turn out to be fundamental and achieve a very important role in this disease.

The used method seems valid and the results seem very clear, although comprehension may not be so easy at reading.

Author Response

We thank the reviewer for his comments on the manuscript. The manuscript is data heavy, so at times may be hard to comprehend for someone not working in the area initially. We hope with the latest revision the reviewer will find the manuscript easy to read.  For a lay person’s comprehension, the main points of the manuscript have been depicted in Figure 13.